# Gut Dysbiosis in Infertile Patients with Persistent Male Accessory Gland Infection

**DOI:** 10.3390/life15060894

**Published:** 2025-05-31

**Authors:** Giuseppe Grande, Andrea Graziani, Raffaele Scafa, Luca De Toni, Andrea Garolla, Alberto Ferlin

**Affiliations:** 1Unit of Andrology and Reproductive Medicine, Department of Systems Medicine, University Hospital of Padova, 35100 Padova, Italy; andrea.garolla@unipd.it (A.G.); alberto.ferlin@unipd.it (A.F.); 2Department of Medicine, University of Padua, 35131 Padua, Italy; andrea.graziani.3@phd.unipd.it (A.G.); raffaele.scafa@gmail.com (R.S.); luca.detoni@unipd.it (L.D.T.)

**Keywords:** prostatitis, male infertility, male accessory gland infection, MAGI, accessory gland

## Abstract

Male tract infections (MTIs) are a common clinical condition, often presenting without any signs nor symptoms of disease. As advised by the European Urology Guidelines dealing with this topic, patients are typically treated with antibiotics alone. Nevertheless, in between 40% and 50% of cases, antibiotic therapy is not effective in eradicating the semen infection. Therefore, persistent semen infection is frequently found upon semen culture evaluation following antibiotic therapy. In this study, we aimed to analyze the fecal microbiota of male infertile patients with persistent MTI in order to verify the prevalence of gut dysbiosis in these patients. We therefore enrolled 20 infertile patients with persistent MTIs after a proper cycle of antibiotic treatment. All patients performed the study for gut microbiota analysis after about 30 days after the last dose of antibiotic treatment. Gut microbiota analysis revealed that 50% of patients with persistent MTI presented a reduction in microbial biodiversity. Indeed, a situation of gut dysbiosis was reported in 75% of patients. In details, the Firmicutes–Bacteroidetes ratio was reduced in 70% of such patients, including 40% of patients where a severe reduction was observed due to an elevated abundance of Bacteroidetes (putrefactive dysbiosis). The most frequent enterotype was Prevotella-dominant (43%). We demonstrated for the first time that patients with recurrent MTIs have enterotypes associated with increased gut permeability and systemic inflammation. Further studies are required to analyze the molecular machinery by which gut dysbiosis exerts its role in patients with MTIs, in particular persistent MTIs, and how supplementation with probiotics might impact in terms of restoring eubiosis, in terms of eradicating the infection, and reducing prostate inflammation and eventually in terms of improving semen evaluation in male infertile patients.

## 1. Background

Infertility is characterized by the inability for a couple to conceive following a minimum of 12 months of consistent, unprotected sexual activity [1]. It is a global health issue that affects millions of individuals and couples worldwide, with profound clinical, psychological, social, and economic consequences. The World Health Organization (WHO) has emphasized the widespread impact of infertility in its recent report, “Infertility prevalence estimates, 1990–2021” published in 2023 [2]. This report highlights the growing prevalence of infertility and underscores the need for improved strategies to address this public health problem. Infertility can be caused by both female and male factors, with male factor infertility (MFI) being a prevalent contributor. Approximately 1 in 20 males experience reproductive issues, making MFI a significant concern in reproductive medicine. Current estimates suggest that MFI, either alone or in combination with female factor infertility, accounts for 30 to 50% of all infertility cases [3,4]. The causes of MFI are diverse, ranging from genetic abnormalities, hormonal imbalances, lifestyle factors, varicocele, and so on.

Among the various causes of MFI, male tract infections (MTIs) are recognized as a significant contributing factor, potentially responsible for up to 15% of male fertility cases [5]. Given the substantial prevalence of MTIs, their timely diagnosis and appropriate treatment are essential aspects that need to be evaluated when managing patients with MFI.

Despite their potentially detrimental impact on male fertility, MTIs often remain underdiagnosed and undertreated, primarily due to their frequent asymptomatic nature. Indeed, these infections can be present in completely asymptomatic males as well as in those experiencing clinical symptoms such as pain, discomfort, or abnormal semen parameters. A variety of bacterial, viral, and fungal pathogens, including human papillomavirus (HPV), can infect the male seminal tract and have been found to negatively impact fertility, either directly or indirectly [6,7].

As previously mentioned, MTIs frequently present without clinical symptoms, making them extremely difficult to detect without thorough clinical evaluation. The etiology of these infections, including the type of pathogen involved, the course of the disease, and the specific area of the male reproductive tract affected, are all factors that can significantly influence the fertility potential and overall reproductive outcomes.

MTIs can impair male fertility potential through several biological mechanisms, which include direct damage to sperm cells, inflammatory immune responses, and structural obstructions within the reproductive tract. For example, the infection-related inflammatory process can alter the secretory activity of accessory glands, leading to changes in seminal plasma composition and impairing sperm survival [8]. Additionally, it may also lead to a structural obstruction of the ejaculatory ducts due to edema or fibrosis of the seminal tract [9]. Furthermore, another consequence of the inflammatory microenvironment caused by MTIs is oxidative stress which may lead to the dysregulation of spermatogenesis, compromising sperm quality and function. Ultimately, this can result in a situation of reduced fertility potential and, therefore, MFI [10,11,12].

As recommended by the European Urology guidelines, antibiotic therapy is the first-line treatment for bacterial MTIs [13]. However, the choice of antibiotic should be carefully tailored based on the type of infection, the patient’s clinical characteristics (such as drug allergies or the presence of kidney failure), as well as the results of antimicrobial susceptibility testing to ensure optimal therapeutic outcomes.

Among available antibiotics, fluoroquinolones are considered the gold standard for the treatment of chronic bacterial prostatitis due to their excellent pharmacokinetic properties. These antibiotics demonstrate high bioavailability, effective penetration into the prostate, with outstanding activity against both common and uncommon bacterial pathogens. Moreover, fluoroquinolones have a favorable safety profile [14,15]. Since only lipid-soluble, low-molecular-weight medications that are not closely associated with plasma proteins can pass through the epithelial barrier, high dosages and long-term therapy are typically recommended despite potential side effects [14,16].

Nevertheless, antibiotic therapy is not always successful. Studies indicate that, in 40% to 50% of cases, antibiotic therapy fails to completely eradicate the infection, as persistent infection is often detected in semen cultures following the end of therapy [17].

Given the limitations of antibiotic therapy, there is a growing interest in finding adjunctive or alternative therapeutic strategies for MTIs. A promising field of research involves the role of probiotics in improving the clearance of persistent infections. A previous study evaluating 104 patients with MFI demonstrated that probiotic treatment might enhance the efficacy of antibiotic therapy in eliminating persistent infections [18], thus supporting the hypothesis that the modulation of gut microbiota might play a role in the treatment of patients with male tract persistent infections. In particular, in this study, 84 patients were treated with *Enterococcus faecium* and *Saccharomyces boulardii* together with antibiotics, followed by a two-week treatment with Lactobacilli. Results showed that only 10 out of 20 patients (50%) who received antibiotics alone reported negative microbiological analysis post-treatment. In contrast, 64 out of 84 patients (76.2%) who received probiotics alongside antibiotics achieved complete eradication of infection, as confirmed by negative semen cultures and the microbiological studies of prostatic secretion after the end of treatments (*p* < 0.05).

Given these emerging insights, the interplay between gut microbiota and male reproductive health warrants further investigation. Gut dysbiosis—an imbalance in the composition of the intestinal microbiota—has been implicated in various systemic diseases, and its potential role in MFI remains an emerging area of interest. Based on these premises, our study aimed to analyze the fecal microbiota of patients with persistent male tract infections to assess the prevalence of gut dysbiosis in these patients. Understanding the potential interactions between gut microbiota, systemic inflammation, and male reproductive function could pave the way for the novel therapeutic strategies of MFI related to MTIs.

## 2. Materials and Methods

We present a case series collected in a third-level infertility clinic at the University Hospital of Padova, Padova, Italy, between June 2022 and December 2023.

The study protocol followed the standard clinical approach and the principles outlined in the Declaration of Helsinki. Informed consent to collect the data anonymously for scientific purposes was obtained from the study participants. IRB approval was requested.

The design of the study has been reported in Figure 1.

We included a group of individuals who are between the ages of 20 and 55 and seeking treatment for primary MFI, with testicular volume in the normal range (12–25 cc for right testis, 11–12 cc for left testis) and normal follicle-stimulating hormone (FSH) concentrations (<8 mUI/L), with the microbiological analysis of semen and/or prostatic secretions that revealed the presence of Gram-negative bacteria (defined as the growth of more than 10^3^ CFU/mL of pathogenic bacteria in cultures of diluted seminal plasma and/or secretions obtained after prostatic massage) and with scrotal ultrasound (US) and transrectal ultrasound (TRUS) signs of prostate–vesicular–epididymitis (PVE).

History of cryptorchidism, orchitis, testicular trauma or torsion, hypogonadism, exposure to chemicals at work, Y chromosome microdeletions, karyotype abnormalities, Cystic Fibrosis Transmembrane Conductance Regulator (CFTR) mutations, reduced testicular volumes, FSH levels greater than 8 mUI/L, fever, drug use (except for antibiotics) within three months before study enrollment, and azoospermia were among the exclusion criteria.

Every patient received a thorough physical and andrological examination, a standard semen assay, and a full microbiological study, which included a culture for atypical pathogens such as Mycoplasmas and Chlamydia in semen and in prostatic secretion after prostate massage, and transrectal prostatic ultrasound evaluation for prostate and male semen tract. The International Prostate Symptom Score (IPSS), which has been validated in Italian [18], was administered to each patient and self-completed.

All patients received antibiotic treatment according to the antibiogram, generally using fluoroquinolones (either 500 mg of levofloxacin or 1 g of ciprofloxacin every day for 14 days). Patients with infection by *Ureaplasma urealyticum* received doxiciclin 100 mg daily for 10 days.

Following treatment, a thorough microbiological analysis was conducted again after 2 weeks.

We only included in the present study patients with persistent infection (the same bacteria) after antibiotic treatment. We therefore enrolled in the present study 20 patients with persistent male tract infection by the same germ after one cycle of antibiotic treatment.

All patients with persistent male tract infection performed the study for gut microbiota analysis after almost 30 days after the last dose of antibiotic treatment.

Following the manufacturer’s instructions, the QIAmp DNA stool kit was used to extract DNA from each sample, in accordance with gut microbiota analysis protocols (Qiagen Ltd., Strasse, Germany). Before being employed for 16S rRNA studies, extracted DNA samples were stored at −20 °C. Using the primer pair Probio_Uni (5-CCTACGGGRSGCAGCAG-3′)/Probio_Rev (5-ATTACCGCGGCTGCT-3′), which targets the V3 region of the 16S rRNA gene sequence, partial 16S rRNA gene sequences were amplified from extracted DNA. The incomplete 16S rRNA gene-specific amplicons were then further analyzed using the 16S Metagenomic Sequencing Library Preparation Protocol (Part #15044223 Rev. B; Illumina, San Diego, CA, USA) after the Illumina adaptor overhang nucleotide sequences were inserted. A VeritiTM Thermocycler (Applied Biosystems; Foster City, CA, USA) was used to perform the amplifications.

Electrophoresis on a 2200 TapeStation Instrument (Agilent Technologies; Santa Clara, CA, USA) was used to assess the PCR amplicons’ integrity. Agencourt AMPure XP DNA purification beads (Beckman Coulter Genomics GmbH, Krefeld, Germany) were used in a magnetic purification step to exclude the primer dimers from PCR products that were generated after a segment of the 16S rRNA gene was amplified. The fluorimetric Qubit quantification equipment (Life Technologies; Carlsbad, CA, USA) was used to measure the amplified sequence library’s DNA concentration. To create the pooled final library, amplicons were diluted to 4 nM and 5 μL of each diluted DNA amplicon sample was combined.

An Illumina MiSeq sequencer equipped with the MiSeq Reagent Kit v3 chemicals-600 cycles (Illumina Inc., San Diego, CA, USA) was used to perform paired-end sequencing (250 bp × 2). As previously mentioned [16], QIIME2 [15] was used to parse the FASTQ data. After merging paired-end reads, quality control was implemented, allowing sequences that range in length from 140 to 400 base pairs, a mean sequence quality score greater than 25, and the ability to truncate a sequence at the first base if a low-quality sequence was discovered within a rolling 10 bp window. Sequences having forward and/or reverse primer mismatches were left out. DADA2 was used to determine 16S rRNA ASVs (amplicon sequence variants) at 100% sequence homology [17], and those with less than 10 sequences were filtered out.

The relative abundance of each taxonomic category for every sample was determined by analyzing the biological observation matrix (BIOM) using the summarize_taxa.py script. Using a reference dataset from the SILVA database v. 132 and QIIME2, all readings were categorized to the lowest taxonomic rank [18]. The QIIME2 2024.5 software suite’s alpha_rarefaction.py script was used to assess the samples’ microbial richness (α-diversity) with default settings. After sequencing, a bespoke script based on the QIIME2 software package was used to parse the FASTQ data. To rebuild the whole Probio-bif_Uni/Probio-bif_Rev amplicons, paired-end reads were put together. Sequences with homopolymers longer than 7 bp and primers that did not match were eliminated, but quality control kept sequences that were of 100–400 bp and had mean sequence quality scores greater than 20.

Since, in our study, we did not enroll a control group, we decided to compare the results obtained with the results detectable by the control database provided by the laboratory (MyMicrobiota, Cremona, Italy) and reporting data from 36,000 healthy subjects, that provided us with the results of the analyses. The control alpha-biodiversity, intended as the number of ASVs, is 153 ± 45, and the Firmicutes–Bacteroidetes ratio is 0.8.

The data were presented as mean ± standard deviation (SD). Statistical analysis employed either an unpaired two-tailed Student *t*-test. Significance was considered at *p* < 0.05. The statistical software utilized for these analyses was GraphPad Prism 10.3.1.

## 3. Results

Clinical, seminal, and hormonal characteristics of the patients have been reported in Table 1.

Semen culture revealed persistent male tract infection due to the pathogens reported in Table 2. All patients presented a mono-infection and no patients reported infection by more than one germ.

Gut microbiota analysis showed how 50% of patients with persistent MTI presented a reduction in microbial biodiversity. In detail, biodiversity, evaluated by the index Observed OTUs, was 88 ± 34, which was significantly reduced as compared to the one reported in the reference group of 36,000 healthy subjects (153 ± 45, *p* < 0.05).

In particular, a situation of gut dysbiosis has been reported in 75% of patients. The ratio Firmicutes–Bacteroidetes was reduced in 70% of these patients, including a 40% in which a severe reduction was observed, due to an elevated abundance of Bacteroidetes (putrefactive dysbiosis).

In detail, the most frequent enterotype was Prevotella-dominant (43%). In 28% of patients, an increase in Bacteroidetes has been reported.

## 4. Discussion

The word “microbiota” first appeared at the beginning of the 20th century and has since become a fundamental concept in medical research. The term refers to the collection of microorganisms, including bacteria, yeasts, and viruses, that reside in several organ systems of the human body (i.e., gastrointestinal tract, skin, lung, and oral cavity) [19]. These microbial communities play a crucial role influencing a variety of physiological processes, including digestion, immune function, and protection against pathogenic microbes. Additionally, human microbiota—sometimes referred as “the hidden organ”—contributes almost 150 times as much genetic information as the whole human genome [20]. Despite their frequent interchange, the words “microbiota” and “microbiome” are not synonymous, and a clear distinction exists between them. If the word “microbiome” describes the collection of all environmental microorganisms’ genomes, on the other hand, the word “microbiota” describes the live bacteria that live in a particular habitat [21].

Even though an association between illnesses/clinical diseases and gut microbiome was first postulated in the 20th century, little was known about the gut microbiome due to the limited microbiological techniques available at the time, which included the sole microbiological culture, which hindered a comprehensive understanding. However, in recent decades, recent advances in molecular biology and sequencing technologies have revolutionized microbiome research. Technologies for next-generation sequencing (NGS), particularly those that seek to target 16S-seq amplicon sequencing, enable the identification and measurement of microorganisms that exist in humans to previously unheard-of levels of resolution [22], allowing for new information on how the microbiome affects health and illness.

One of the most important populations of microorganisms in the human body is the gut microbiota, mostly composed of bacterial cells, which consist of an estimated 10^13^ bacterial cells. Our gut’s microbiota is probably a pivotal component in preserving our health [23].

We clearly demonstrated in our study a high prevalence of gut dysbiosis among patients with persistent MTIs. Several biological mechanisms have been proposed to explain the relationship between gut dysbiosis and male accessory gland infection/inflammation and with MFI [24,25]. In detail, one such mechanism could be the compromise of the gut barrier due to gut microbiota dysregulation, which leads to increased intestinal permeability and to “leaky gut syndrome”. The translocation of the gut microbiota and its metabolites into the systemic circulation trigger immune activation and the release of pro-inflammatory factors which induce a chronic inflammation. Prostate inflammation might therefore potentially result from bacterial components (like lipopolysaccharides) or gut microbiota metabolites (like the short chain fatty acid, SCFAs) that reach the systemic circulation through the venous and lymphatic routes.

Thus, physiologically active metabolites, including bile acid derivatives, SCFAs, and tryptophan metabolites, are produced by the gut microbiota and engage in a number of pathways and mechanisms that impact the host’s physiological and pathological processes [26]. Among the most significant metabolites of the gut microbiota there are the SCFAs, consisting of butyrate, propionate, and acetate. They are produced by the fermentation of food fibers and the breakdown by the gut flora. In particular, firmicutes produce higher quantities of butyrate [27]. Butyrate has been proven in several studies to have anti-inflammatory properties because it inhibits the activation of NF-κB [25]. Additional research also demonstrated that butyrate can activate PPAR-γ [28] and inhibit IFN-γ signaling [29], thus amplifying its anti-inflammatory action. Beyond its role in inflammation, butyrate is also known to enhance the integrity of the intestinal epithelial barrier, reinforcing its protective function and reducing gut permeability [30].

Therefore, the reduction in Firmicutes observed in the present study in patients with recurrent male tract infection might play a role in the negative modulation of systemic inflammatory systems, including in the prostate, and in increasing gut permeability.

We, moreover, observed that the Prevotella-dominant enterotype was the most common enterotype in our patient cohort. This finding aligns with previous studies showing a strong correlation between Prevotella overgrowth and immune system overactivation [31,32]. Elevated Prevotella levels have been associated with various inflammatory conditions, including inflammatory bowel diseases [33] and rheumatoid arthritis [33]. The presence of excessive Prevotella species might contribute to aberrant immune response activation in the gut and might interfere with the intestinal barrier’s normal function [34].

In conclusion, the specific enterotypes reported in our population (Prevotella-dominant and Bacteroidetes-dominant) are associated with increased gut permeability and systemic inflammation. There is growing evidence that the gut microbiota plays a part in urologic health, even when dealing with male tract and MFI, potentially via its impact on intestinal permeability [35]. Increased intestinal permeability permits intestinal lumen fluid, leukocytes, toxins, macromolecules, and other substances that may exacerbate inflammation to seep into the bloodstream [36]. Prostate conditions including inflammation and the development of prostate cancer have been linked to intestinal permeability [37]. Furthermore, previous studies demonstrated that the use of probiotics, aimed at normalizing gut microbiota, might have clinical significance in patients with persistent MTIs, thus increasing the eradication rate [18].

Finally, the evaluation of gut microbiota might play a role in patients with MFI. In particular, in light of the evidence that a “symmetric” evaluation is suggested in female infertile patients [38], the evaluation of microbiota in male patients might be suggested in patients with “idiopathic” MFI, in which etiology has not been determined and we may speculate for a putative role of an alteration in gut microbiota [39,40].

However, despite these promising findings, it is important to acknowledge some limitations of our study. First of all, the small sample size limits the statistical power of the study. Furthermore, in the absence of a control group of fertile subjects, we used as comparison a previously analyzed group of healthy subjects. Moreover, diet, previous antibiotic history, lifestyle, and other factors affecting microbiota composition have not been evaluated. Finally, since it is a preliminary study on patients with recurrent infections, we do not have data about gut microbiota before the antibiotic treatment; further studies might be performed to evaluate the presence of gut dysbiosis in patients with male tract infection at diagnosis before antibiotic treatment, and the effect of treatments (antibiotic and probiotic/prebiotic treatments) on gut microbiota.

Further studies are required in order to delve into the molecular machinery by which gut dysbiosis exerts its role in patients with MTIs and how supplementation with probiotics might impact in terms of restoring eubiosis, in terms of eradicating the infection and reducing prostate inflammation, and eventually in terms of improving semen evaluation in male infertile patients.

## Figures and Tables

**Figure 1 life-15-00894-f001:**
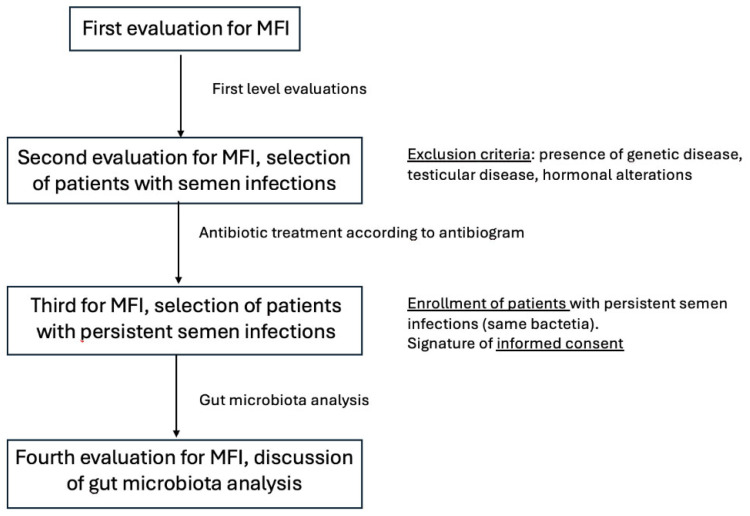
Design of the study.

**Table 1 life-15-00894-t001:** Clinical, seminal, and hormonal characteristics of the studied population. Abbreviations: FSH: follicular stimulating hormone; LH: luteinizing hormone.

	Results (Mean ± SD)
Age (male) [years]	36.7 ± 5.9
Age (female partner) [years]	30.5 ± 3.7
Duration of infertility [years]	1.8 ± 1.1
Right testicular volume	13.9 ± 4.3
Left testicular volume	13.2 ± 4.2
Semen volume [mL]	3.5 ± 1.6
Semen pH	7.9 ± 0.4
Sperm concentration [×10^6^/mL]	11.5 ± 11.1
Total sperm count [×10^6^]	40.8 ± 50.0
Total sperm motility [%]	33.4 ± 23.1
Normal sperm morphology [%]	6.6 ± 5.1
FSH [mUI/mL]	4.4 ± 5.2
LH [mUI/mL]	3.4 ± 5.2
Testosterone [ng/mL]	3.9 ± 4.7

**Table 2 life-15-00894-t002:** Prevalence of the different pathogens isolated in semen in the studied population of patients with resistant male tract infection.

Isolated Pathogen	Prevalence [*n*, (%)]
*Ureaplasma urealyticum*	7/20 (35%)
*Escherichia coli*	5/20 (25%)
*Enterococcus faecalis*	5/20 (25%)
*Citrobacter freundii*	2/20 (10%)
*Staphylococcus*	1/20 (5%)

## Data Availability

The data presented in this study are available on request from the corresponding author. The data are not publicly available due to privacy/ethical restrictions.

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
