# Peer review of "Gut Dysbiosis in Infertile Patients with Persistent Male Accessory Gland Infection"

_life, 2025, doi:10.3390/life15060894_

Round 1
Reviewer 1 Report
Comments and Suggestions for Authors
The authors present the results of their study focused on the analysis of the gut microbiota of infertile males with persistent MTI. This topic is of interest to the audience and aligns with the main goals of Life.
The manuscript was reviewed using the Reporting guidelines https://www.equator-network.org/reporting-guidelines/ and the comments by sections are as follows:
The rationale for the study in the context of existing knowledge and the study objectives are presented in the Introduction as required.
There are some issues with reporting of Subjects and Methods. Consider identifying the study design, setting, and period of data collection.Please explain howw the study size was calculated.
The exclusion criteria - “drug use within three months before study enrollment” (Lines 97-98)- could be clarified; in current version this criteria doesn’t correspond with antibiotic intake: “We therefore enrolled in the present study 20 patients with persistent male tract infection after antibiotic treatment” (Lines 113-114). Consider to add flow diagram and clarify when exactly the patients signed the Informed consent – before or after antibiotics intake.
Regarding the phrase (Lines 105-107): All patients with symptoms or signs of inflammation in semen analysis and/or TRUS, and infection documented at microbiological analysis received antibiotic treatment according to antibiogram. Does it mean that not ALL the participant had the inflammation? How homogeneous was the study group?
Table 2. – Please clarify whether mono infections were present in the study group. No mixed ones?
Unfortunately, in the current version of the manuscript, there are no data on alpha- biodiversity parameters in the study group. The authors only present the percentage of persons with decreased biodiversity, without any descriptions.
The absence of a control group was a key issue to obtain qualitative results and it explains a relatively low scientific significance of the manuscript in it's current version.
Author Response
The authors present the results of their study focused on the analysis of the gut microbiota of infertile males with persistent MTI. This topic is of interest to the audience and aligns with the main goals of Life.
The manuscript was reviewed using the Reporting guidelines https://www.equator-network.org/reporting-guidelines/ and the comments by sections are as follows:
The rationale for the study in the context of existing knowledge and the study objectives are presented in the Introduction as required.
We thank the Reviewer.
There are some issues with reporting of Subjects and Methods. Consider identifying the study design, setting, and period of data collection. Please explain how the study size was calculated.
Since the reduced study size, we clearly reported among the limits of the study, we reported that this is a case series of patients with persistent MTI.
The exclusion criteria - “drug use within three months before study enrollment” (Lines 97-98)- could be clarified; in current version this criteria doesn’t correspond with antibiotic intake: “We therefore enrolled in the present study 20 patients with persistent male tract infection after antibiotic treatment” (Lines 113-114). Consider to add flow diagram and clarify when exactly the patients signed the Informed consent – before or after antibiotics intake.
We added a flow diagram of the study design as requested and clarified that exclusion criteria was for any previous drug use, except for antibiotics.
Regarding the phrase (Lines 105-107): All patients with symptoms or signs of inflammation in semen analysis and/or TRUS, and infection documented at microbiological analysis received antibiotic treatment according to antibiogram. Does it mean that not ALL the participant had the inflammation? How homogeneous was the study group?
Thank you for your comment. All patients who have been enrolled received the treatment. We have modified the sentence to avoid any possible misunderstanding.
Table 2. – Please clarify whether mono infections were present in the study group. No mixed ones?
We have stated that “All patients presented a mono-infection and no patients reported infection by more than one germ.”
Unfortunately, in the current version of the manuscript, there are no data on alpha- biodiversity parameters in the study group. The authors only present the percentage of persons with decreased biodiversity, without any descriptions.
We have reported the biodiversity index, evalutated by the index Observed OTUs, and compared it with the reference group of 36,000 healthy subjects, thus underlining the reduction on biodiversity.
The absence of a control group was a key issue to obtain qualitative results and it explains a relatively low scientific significance of the manuscript in it's current version.
We understand that the absence of control group is a limit of the study; however microbiota analysis is performed intrinsically by a comparison with the control group of healthy subjects. We therefore compared the results with the control database provided by the laboratory (MyMicrobiota, Cremona; Italy) and reporting data from 36,000 healthy subjects, that provided us with the results of the analyses. We clearly reported this point in the method section and among the limits of the study.
Reviewer 2 Report
Comments and Suggestions for Authors
Dear authors, after reading your paper, I think there are some drawback and inconsistencies
- The major drawback is the lack of a control group
- If they had persistent infection, how many times they received antibiotic? Do you think that the prolonged treatment could influence the results?
- Have the patients received probiotic supplements while taking antibiotics?
- In case of persistent infection, the culture showed the same germ? Did they receive the same treatment?
Author Response
Dear authors, after reading your paper, I think there are some drawback and inconsistencies. The major drawback is the lack of a control group
We thank the Reviewer for his/her kind observations. We understand that the absence of control group is a limit of the study; however microbiota analysis is performed intrinsically by a comparison with the control group of healthy subjects. We therefore compared the results with the control database provided by the laboratory (MyMicrobiota, Cremona; Italy) and reporting data from 36,000 healthy subjects, that provided us with the results of the analyses. We clearly reported this point in the method section and among the limits of the study.
If they had persistent infection, how many times they received antibiotic? Do you think that the prolonged treatment could influence the results?
We have clearly stated that we have enrolled patients with persistent male tract infection by the same germ after one cycle of antibiotic treatment. To evaluate the effect of antibiotic treatment on gut microbiota we should evaluate gut microbiota before the first antibiotic treatment, and this is an interesting perspective for future researches. We have clearly discussed this point among the limits of the study.
Have the patients received probiotic supplements while taking antibiotics?
Patients received antibiotic treatment alone, not associated with probiotics, accordingly to current guidelines. Further studies are therefore needed to evaluate the effect of association schemes with probiotic/prebiotic treatment on gut microbiota in these patients.
In case of persistent infection, the culture showed the same germ? Did they receive the same treatment?
As clearly stated, we performed the study after one cycle of antibiotic treatment in patients presenting a culture positive for the same germ.
Reviewer 3 Report
Comments and Suggestions for Authors
-
The study includes only 20 patients, which limits its statistical power. A larger cohort would be necessary to validate the findings. You should discuss this limitation in more detail and, if possible, increase the sample size.
-
The study compares patient microbiota data with a reference database instead of a matched healthy control group. A direct comparison with fertile men without MTIs would strengthen the validity of the results.
-
Diet, antibiotic history, lifestyle, and other factors affecting microbiota composition are not accounted for. The authors should discuss potential confounders and limitations.
-
While the study presents changes in microbiota composition, no statistical tests (e.g., p-values, and confidence intervals) are explicitly stated. Statistical validation of observed differences would enhance credibility.
-
Line 13: "Male tract infections (MTIs) fare a common clinical condition…" → should be "are a common clinical condition".
-
Lines 25-26: "including a 40% of patients in which a severe reduction was observed…" → should be "including 40% of patients where a severe reduction was observed…".
-
Line 56: "As advised by the European Urology guidelines, patients are typically treated with antibiotics alone" → should be "with antibiotics as the primary treatment."
-
Lines 237-238: "The enterotypes of microorganisms of gut microbiota – that we reported in our population - are associated with increased gut permeability and systemic inflammation." → This sentence is unclear and should be revised for better readability.
-
Figure 1 (Lines 180-182): The figure caption should provide more detail. What does "Mixed Dysbiosis" represent? Clarify the biological significance.
-
Tables (Lines 166-174): The manuscript presents useful clinical data, but it lacks statistical comparisons. Including p-values would make the findings more robust.
Author Response
The study includes only 20 patients, which limits its statistical power. A larger cohort would be necessary to validate the findings. You should discuss this limitation in more detail and, if possible, increase the sample size.
We thank the Reviewer for his/her kind observations. We clearly discussed the point among the limits of the study. However we decided to include in the study an homogeneous group of patients with infertility and recurrent male tract infection after antibiotic treatment by the same germ. This choice means that it is not very common to observe these patients in clinical practice. Therefore, although the sample size analyzed was small, its homogeneity allowed us to obtain interesting preliminary data, which will obviously need to be confirmed on larger populations.
The study compares patient microbiota data with a reference database instead of a matched healthy control group. A direct comparison with fertile men without MTIs would strengthen the validity of the results.
We understand that the absence of control group is a limit of the study; however microbiota analysis is performed intrinsically by a comparison with the control group of healthy subjects. We therefore compared the results with the control database provided by the laboratory (MyMicrobiota, Cremona; Italy) and reporting data from 36,000 healthy subjects, that provided us with the results of the analyses. We clearly reported this point in the method section and among the limits of the study.
Diet, antibiotic history, lifestyle, and other factors affecting microbiota composition are not accounted for. The authors should discuss potential confounders and limitations.
We discussed these potential confounders and limitations.
While the study presents changes in microbiota composition, no statistical tests (e.g., p-values, and confidence intervals) are explicitly stated. Statistical validation of observed differences would enhance credibility.
We reported quantitative data on microbiota composition and performed statistical tests (p-value <0.05)
Line 13: "Male tract infections (MTIs) fare a common clinical condition…" → should be "are a common clinical condition".
Lines 25-26: "including a 40% of patients in which a severe reduction was observed…" → should be "including 40% of patients where a severe reduction was observed…".
Line 56: "As advised by the European Urology guidelines, patients are typically treated with antibiotics alone" → should be "with antibiotics as the primary treatment."
Lines 237-238: "The enterotypes of microorganisms of gut microbiota – that we reported in our population - are associated with increased gut permeability and systemic inflammation." → This sentence is unclear and should be revised for better readability.
We modified the manuscript according to the Reviewer’s suggestions.
Figure 1 (Lines 180-182): The figure caption should provide more detail. What does "Mixed Dysbiosis" represent? Clarify the biological significance.
We have modified the manuscript and deleted the figure, reporting in the text the 2 major enterotypes (Prevotella- and Bacteroidetes- dominant)
Tables (Lines 166-174): The manuscript presents useful clinical data, but it lacks statistical comparisons. Including p-values would make the findings more robust.
We performed statistical analysis and included p-values in the comparison of biodiversity in patients and healthy controls.
Round 2
Reviewer 1 Report
Comments and Suggestions for Authors
The authors edited the article and made all recommended corrections.
Reviewer 3 Report
Comments and Suggestions for Authors
You have made a commendable effort in enhancing the quality of your manuscript.